# Extracellular Vesicles in Triple–Negative Breast Cancer: Immune Regulation, Biomarkers, and Immunotherapeutic Potential

**DOI:** 10.3390/cancers15194879

**Published:** 2023-10-07

**Authors:** Kaushik Das, Subhojit Paul, Arnab Ghosh, Saurabh Gupta, Tanmoy Mukherjee, Prem Shankar, Anshul Sharma, Shiva Keshava, Subhash C. Chauhan, Vivek Kumar Kashyap, Deepak Parashar

**Affiliations:** 1Department of Cellular and Molecular Biology, The University of Texas at Tyler Health Science Center, Tyler, TX 75708, USA; shivakeshava.gaddam@uttyler.edu; 2School of Biological Sciences, Indian Association for the Cultivation of Science, Jadavpur, Kolkata 700012, India; bcsp3@iacs.res.in (S.P.); bcag3@iacs.res.in (A.G.); 3Department of Biotechnology, GLA University, Mathura 281406, India; saurabh.gupta@gla.ac.in; 4School of Medicine, The University of Texas at Tyler Health Science Center, Tyler, TX 75708, USA; tanmoy.mukherjee@uttyler.edu; 5Department of Neurobiology, The University of Texas Medical Branch, Galveston, TX 77555, USA or premshankar.1506@gmail.com; 6Division of Hematology & Oncology, Department of Medicine, Medical College of Wisconsin, Milwaukee, WI 53226, USA; 7Department of Immunology and Microbiology, School of Medicine, University of Texas Rio Grande Valley, McAllen, TX 78504, USA; subhash.chauhan@utrgv.edu (S.C.C.); vivek.kashyap@utrgv.edu (V.K.K.); 8South Texas Center of Excellence in Cancer Research, School of Medicine, University of Texas Rio Grande Valley, McAllen, TX 78504, USA

**Keywords:** extracellular vesicles, classification, breast cancer subtypes, triple negative breast cancer, biomarker, immune regulation, cancer progression, immunotherapy

## Abstract

**Simple Summary:**

Triple–negative breast cancer (TNBC), an aggressive phenotype, is commonly attributed to the loss of estrogen receptor, progesterone receptor, and human epidermal growth factor receptor, thereby posing unique challenges in treatment via conventional targeted therapies. Although present chemotherapeutic regimens show promise against TNBC morbidity; the variability in treatment outcomes among patients and emerging resistance limit their potential. Moreover, due to significant mutational burden, TNBC is considered as highly immunogenic. Extracellular vesicles (EVs) have shown to regulate multiple human pathologies including cancer due to their role in transferring bioactive molecules between cells, evading immune surveillance. Conversely, the biological novelties of EVs are also exploited to develop experimental therapies. Here, we review studies in which EVs contribute to the progression of human TNBC and their immune regulation. Understanding these mechanistic details may open up avenues for repurposing EV–based immunotherapeutic strategies in the context of human TNBC treatment.

**Abstract:**

Triple–negative breast cancer (TNBC) is an aggressive subtype accounting for ~10–20% of all human BC and is characterized by the absence of estrogen receptor (ER), progesterone receptor (PR), and human epidermal growth factor receptor 2 (HER2) amplification. Owing to its unique molecular profile and limited targeted therapies, TNBC treatment poses significant challenges. Unlike other BC subtypes, TNBC lacks specific molecular targets, rendering endocrine therapies and HER2–targeted treatments ineffective. The chemotherapeutic regimen is the predominant systemic treatment modality for TNBC in current clinical practice. However, the efficacy of chemotherapy in TNBC is variable, with response rates varying between a wide range of patients, and the emerging resistance further adds to the difficulties. Furthermore, TNBC exhibits a higher mutational burden and is acknowledged as the most immunogenic of all BC subtypes. Consequently, the application of immune checkpoint inhibition has been investigated in TNBC, yielding promising outcomes. Recent evidence identified extracellular vesicles (EVs) as an important contributor in the context of TNBC immunotherapy. In view of the extraordinary ability of EVs to transfer bioactive molecules, such as proteins, lipids, DNA, mRNAs, and small miRNAs, between the cells, EVs are considered a promising diagnostic biomarker and novel drug delivery system among the prospects for immunotherapy. The present review provides an in–depth understanding of how EVs influence TNBC progression, its immune regulation, and their contribution as a predictive biomarker for TNBC. The final part of the review focuses on the recent key advances in immunotherapeutic strategies for better understanding the complex interplay between EVs and the immune system in TNBC and further developing EV–based targeted immunotherapies.

## 1. Introduction

### 1.1. Extracellular Vesicles: More Than Cellular Dust

Extracellular vesicles (EVs) were first identified in 1946 by Chargaff and West as platelet–derived procoagulant particles in plasma [1]. More than two decades later, Wolf renamed these procoagulant plasma–derived bodies “platelet dust” [2]. A further decade passed before the intriguing discovery of Harding et al. [3] that plasma particles serve as an important mode of cell–cell communication via autocrine, paracrine, or endocrine mechanisms via the transfer of bioactive molecules [4]. In the late 2000s, the advancement of EV research further established the biological implications of EVs [5,6]. In the last decade, EVs have come up as an important contributor in various pathophysiological conditions [7,8,9,10,11,12,13,14], basically via passing on bioactive molecules in the form of DNA, RNAs, miRNAs, proteins, lipids, etc., between the cells [15].

EVs are membrane–enclosed nanoparticles that are released from almost every cell type into the extracellular milieu [4,16]. They are abundantly found in body fluids like blood, saliva, urine, and amniotic fluid, and their presence is detectable within the interstitial spaces between the cells [17,18,19]. Sometimes, the number and composition of EVs are significantly altered in various disease conditions, often rendering them novel biomarkers for different pathophysiological settings [20,21]. However, depending on their biogenesis, release mechanisms, and functions, the EVs are broadly classified into microparticles, exosomes, and apoptotic bodies.

### 1.2. Microparticles

Microparticles (MPs), microvesicles (MVs), or ectosomes are produced by the external budding of the plasma membrane [14]. Initially, the loss of membrane asymmetry results in the externalization of phosphatidylserine (PS), which usually resides in the inner leaflet of the plasma membrane, to the outer leaflet and, hence, it is enriched on the MPs’ surfaces. This is followed by the reorganization of cytoskeletal components to induce membrane curvature, ultimately leading to the liberation of the MPs outside the cells [22,23,24]. Therefore, cytosolic and membrane–associated proteins, such as tetraspanins, which are the proteins clustered at the plasma membrane, are enriched into the MPs, which often serve as an MP marker, regardless of the originating cells [25,26]. However, cytoskeleton–associated proteins, like heat shock proteins (HSPs), integrins, and proteins associated with post–translational modifications (phosphorylation, glycosylation, etc.), are also abundantly found in MPs [27,28,29]. MPs appear to be larger at ~100–1000 nm in diameter [14,19].

### 1.3. Exosomes

Exosomes, on the other hand, are 30–150 nm in diameter [30] and are basically of endocytic origin [14]. Additionally, exosomes are nanovesicles that facilitate cell–to–cell communication, altering the tumor microenvironment through paracrine signaling [31]. Exosomes play an important role in promoting the EMT phenotype and increasing chemoresistance [32]. Conventionally, early endosomal membranes invaginate to produce such exosomes, which mature into multivesicular bodies (MVBs) [14]. Further, these MVBs fuse with the plasma membrane to shed the exosomes outside the cell [14]. The endosomal sorting complexes required for the transport (ESCRT) pathway play a major role in the biogenesis of exosomes [14], thereby ESCRT proteins, including TSG101, Alix, HSP90β, and HSC70, are abundantly found in the exosomes [33,34], which often serve as exosomal markers. In contrast, exosome biogenesis also involves ESCRT–independent mechanisms, which were shown to be mediated by the sphingolipid ceramide [35].

### 1.4. Autophagic EVs

An increasing body of evidence identified an unconventional mechanism, namely, “autophagy”, which is the lysosomal mechanism of bulk degradation of the dysfunctional and unusable components of the cells [36] that helps cellular metabolism and homeostasis. Autophagy is induced by different factors, such as nutrient starvation and the accumulation of old or damaged materials, such as proteins, lipids, and organelles, and plays an important role in tumor progression or suppression. Autophagy can regulate the biogenesis of the exosomes. During autophagy, the cytoplasmic portion is sequestered by an organelle, namely, the phagophore, resulting in the formation of “autophagosomes” that eventually fuse with the MVBs to form “amphisomes” [37,38]. Therefore, both the endosome, as well as autophagosome markers, for example, LC3 and CD63, are found in amphisomes and also found to be enriched with nucleosomes and cytosolic DNA. During starvation, these endosomes can be recycled with the assistance of Rab11, Rab27, and Rab35 [39].

There are 10 members in the Rab GTPases family that facilitate the autophagic process, with the majority functioning at the autophagosome level [40]. The Rab GTPases maintain intracellular trafficking, regulating the traffic between organelles by recruiting effector proteins. Therefore, Rab GTPases are considered the “master regulators” of intracellular trafficking [41]. During early apoptosis in cells, cells secrete apoptotic exosome–like vesicles (AEVs). Moreover, AEVs biogenesis is proteasome–dependent, while AEVs secrete in a caspase–3–dependent manner. The major difference between apoptotic bodies and AEVs is that AEVs lack apoptotic markers, such as GM130, calreticulin, and tubulin [42]. When the autophagosome fuses with lysosomes, which is facilitated by Rab7, this results in the degradation of their contents. As a result, the cell receives nutrients and energy. Second, amphisomes can also fuse with the plasma membrane, leading to the release of exosomes outside of the cell via exophagy [41]. Both MPs and exosomes play crucial roles, not only in maintaining homeostasis but also in the progression and development of various pathophysiological conditions, for example, cancer, due to the transfer of bioactive cargo [43]. MPs and exosomes are readily taken up by the recipient cells, either via direct fusion with the plasma membrane or an endocytic mechanism; in either case, the vesicles’ contents are released into the cytosol of the recipient cells [44,45,46,47]. Figure 1 briefly summarizes the mechanism of autophagic EVs’ biogenesis and the impact on the tumor microenvironment.

### 1.5. Apoptotic Bodies (ApoBDs)

ApoBDs are membrane blebs that are fairly variable in size and diameter, ranging from 50 nm to 5 µm, and are produced by the cells undergoing apoptosis [48]. The induction of apoptosis produces a significant hydrostatic pressure, which is generated during the contraction of the cells, thereby segregating the plasma membrane from the cytoskeleton, as a result of which the apoptotic bodies are released [49]. However, unlike MPs and exosomes, apoptotic bodies carry cellular organelles, chromatin, and a few glycosylated proteins [48,50,51,52]. Consequently, markers for cellular organelles, such as HSP60 for mitochondria, GRP78 for Golgi and endoplasmic reticulum, and histones for nuclear markers, are in the apoptotic bodies [49]. Additionally, these ApoBDs have external features (“Eat–me” signals, as well as nuclear autoantigens) that induce phagocytosis [48,50,51,52] as schematically depicted in Figure 2.

Table 1 summarizes the size distribution, markers, and biogenetic mechanisms of different forms of EVs.

## 2. EVs and Diseases

Increasing evidence indicates that EVs, due to the transfer of heterogeneous cargos between the cells, often influence various pathophysiological conditions. For example, in the case of a urinary tract infection (UTI), mediated by Gram–negative bacteria–induced sepsis, higher expression of procoagulant tissue factor (TF) is often observed on the plasma EVs, which reflects the hypercoagulative states in UTI patients [60]. On the other hand, activated platelet–derived EVs are shown to develop anticoagulant states [61]. In atherosclerosis, negatively charged phospholipids, such as phosphatidylserine (PS), allow for the uptake of intercellular adhesion molecule 1 (ICAM–1)–bearing plaque–derived EVs by the endothelial cells, thereby facilitating the recruitment of pro–inflammatory cells, leading to plaque progression [62]. Furthermore, the expression of fetuin–A is shown to be significantly upregulated, whereas aquaporin 1 (AQP1) is downregulated in the urinary EVs of acute kidney injury (AKI) patients, and hence, fetuin–A+ and AQP1+ EVs are considered biomarkers for AKI [63]. The upregulation of ten signature miRNA molecules (miR–199a–5p, miR–4745–3p, miR–143–3p, miR–4532, miR–193b–3p, miR–199b–3p, miR–25–3p, miR–199a–3p, miR–629–5p, and miR–6087), as well as the downregulation of another ten (miR–23b–3p, miR–141–3p, miR–10a–5p, miR–200c–3p, miR–98–5p, miR–382–5p, miR–200a–3p, miR–483–3p, miR–483–5p, and miR–3911) are often seen in the human follicular fluid (HFF)–derived EVs of PCOS biomarkers [64]. α–synuclein +EVs are shown to be enriched in the cerebrospinal fluid (CSF) of Parkinson’s disease (PD) patients and contribute to PD progression via facilitating α–synuclein aggregation in healthy cells [65]. In congenital myopathies (CMs), EVs from the differentiating fibroblasts were shown to be released into circulation, which enhances the regeneration of muscles; therefore, an elevated level of circulating EVs is considered a diagnostic biomarker of CM [66]. Moreover, it was observed that with respect to healthy controls, the composition of microbial EVs in the blood, urine, and feces of patients with gastrointestinal tract disease is significantly altered, which also serves as a diagnostic and therapeutic biomarker for the progression of the disease [67]. On numerous occasions, EVs were shown to be involved in the development and progression of various types of cancer. For example, miR–144 expression in the EVs of nasopharyngeal carcinoma (NPC) is significantly upregulated, which promotes the migration, invasion, and angiogenesis of endothelial cells [68]. Another study indicated that hypoxic–lung–cancer–cell–derived EVs are enriched with miR–103a, which targets PTEN in macrophages, leading to M2 polarization via the activation of AKT and STAT3, which further contributes to the release of immunosuppressive and pro–angiogenic factors to promote lung cancer progression [69]. Moreover, hepatocellular carcinoma (HCC)–derived EVs are abundant with miR–3129, which promotes the growth and metastasis of HCC via targeting TXNIP [70]. Furthermore, human colorectal cancer (CRC)–derived EVs facilitate tumor progression and metastasis via miR–25–mediated downregulation of SIRT6 [71]. The following section delineates how EVs contribute to the pathogenesis of human breast cancer, specifically human TNBC, in different ways. This is followed by a brief discussion on how EVs contribute as biomarkers for TNBC. The final part of the review discusses how EVs act as an immunotherapeutic mediator in the treatment of human TNBC.

## 3. Breast Cancer and Its Subtypes

Breast cancer is known as a pathological condition in which the malignant cells in the breast start growing out of control [72,73,74]. Breast cancer is considered the second–leading cause of cancer–related death among women worldwide [75], and in 2022, ~287,850 new cases of invasive breast cancer were predicted to arise in the United States, with a predictable death rate of 43,250 [75].

### 3.1. Breast Cancer Subtypes

However, the modern classification of breast cancer on the basis of primary markers (ER, PR, and HER2) [76,77], the Ki–67 proliferation index [78], and basal markers (EGFR, CK5/6) [79] differentiates breast cancer into luminal A, luminal B, HER2 high, normal–like, basal–like, and claudin low. The luminal subtype of breast tumor involves the inner (luminal) breast epithelial cells [80], which was first discovered in 2000 using ER+ immunohistochemical profiling [81]. Depending on HER2 expression, the luminal subtype is further classified into luminal A and luminal B, luminal A is shown to be HER2−, whereas luminal B is HER2+ [82]. Moreover, luminal B is characterized by a worse prognosis and propagates at a faster rate than luminal A [82]. In contrast with luminal subtypes, the HER2 high subtype is ER− and PR− but HER2+ and is characterized by a higher growth rate with a worse prognosis [83]. Interestingly, the HER2 high breast cancer subtype is shown to be highly sensitive to the chemotherapeutic drug trastuzumab [84]. The normal breast cancer subtype is characterized by the loss of expression of the proliferation gene, which is carried by a low percentage of cancer cells [80]. The basal–like subtype is ER, PR, and HER2 negative (HER2−), and is so named because of the similarity with cytokeratins (CKs) 5/6, and EGFR expression [81,85,86,87,88]. The basal–like breast cancer subtype is also known as TNBC, which was shown to be associated with a mutation in the breast cancer gene 1 (BRCA1) [89]. However, recent evidence indicates that not all TNBC can be considered basal types. Prat et al. identified ~70–80% of all TNBC as basal type [90]. Due to the deficiencies of ER, PR, and HER2 expressions, TNBC cells often show resistance against various hormones and targeted therapy and, therefore, can only be treated with cytotoxic drugs [91]. The claudin–low subtype, sometimes also referred to as TNBC, is characterized by lower expressions of E–cadherin, mucin1, EpCAM, and claudins (3, 4, 7) and limited expression of Ki–67 and luminal markers compared with the other breast cancer subtypes [90]. In contrast, an elevated expression of genes associated with the epithelial–to–mesenchymal transition (EMT), such as vav1, CD79b, and CD14, and cancer stemness are often observed in the claudin–low subtype [92,93,94]. Table 2 briefly illustrates the immunohistochemistry–based subtyping of various breast cancers.

### 3.2. Subtypes of TNBC and Their Treatment Measures

Based on the gene expression profile [95], TNBC is a type of breast cancer that has a lot of different kinds of cells. It can be further broken down into basal–like 1 (BL–1), basal–like 2 (BL–2), immunomodulatory (IM), luminal androgen receptor (LAR), mesenchymal (M), and mesenchymal stem–like (MSL). The BL–1 and BL–2 subtypes have similar expression profiles of cell cycle and cell division–associated genes.

However, BL–1 exerts a relatively higher expression of DNA replication and repair–associated genes, whereas BL–2 shows an elevated expression of growth–factor–signaling genes [95]. The treatment measures for BL–1 include PARP inhibitors [96] and platinum compounds [97,98], whereas the inhibition of growth signaling was shown to be effective against BL–2 [99]. In contrast, the IM subtype often demonstrates enhanced expression of immune–response–related genes, such as genes associated with the NK–cell pathway, Th1/Th2 pathway, B–cell receptor (BCR), antigen processing, and cytokine signaling; therefore, immune checkpoint inhibition is often found to be an effective measure against IM [100]. The LAR subtype, on the other hand, shows an over–expression of genes related to androgen receptors, as well as several hormonally regulated pathways [95,101]. The androgen receptor agonist bicalutamide was shown to be effective in treating the LAR subtype [102]. As the names suggest, both M and MSL are associated with the over–expression of genes related to cell motility, extracellular receptor interaction, and cellular differentiation pathways. However, subtle differences exist, for example, the expression of claudins (3, 4, 7) was shown to be significantly lower in MSL [95], which is often considered claudin–low breast cancer [103]. The treatment measures for the M subtype include targeting the Wnt/β–catenin pathway [104], PI3K/mTOR pathway [95,105], and TGFβ receptor kinase [106]. Figure 3 summarizes the different subtypes of TNBC, their general characteristics, and treatment measures.

## 4. The Role of EVs in the Progression of TNBC

Numerous studies identified the active participation of EVs in the progression of human TNBC; in some instances, EVs are associated with the growth of TNBC, whereas in other cases, they influence the metastatic dissemination of TNBC. Table 3 briefly discusses how EVs contribute to the growth and metastasis of TNBC in different ways.

### 4.1. EVs in TNBC Growth

Due to the extraordinary capability of the EVs to transfer bioactive molecules between the cells, EVs are often shown to promote TNBC growth, thereby contributing to the progression of TNBC. For example, EVs from HCC1806 TNBC cells induce the proliferation of and drug resistance in the normal breast epithelial cells MCF10A by involving PI3K/AKT, MAPK, and HIF1α pathways [107]. Moreover, plant–derived sesquiterpene lactone deoxyelephantopin (DET) and its derivative, namely, DET derivative 35 (DETD–35), were shown to evoke the release of EVs from the human TNBC cells MDA–MB–231, which, in turn, inhibit the proliferation of MDA–MB–231 cells via the down–regulation of cell adhesion, migration, and angiogenesis [108]. Cancer–associated fibroblasts (CAFs) in the tumor microenvironment (TME) often trigger the growth of tumor cells [14]. A recent study showed that integrin β4 (ITGB4) + EVs that are released from MDA–MB–231 cells promote CAFs’ growth by triggering the glycolytic pathway via BNI3PL–dependent mitophagy and lactate production, which, in turn, induce the growth of the TNBC tumor [109]. Furthermore, mesenchymal stem cell (MSC)–derived EVs, via the transfer of miR–106a–5p, promote TNBC tumor progression upon downregulating the expression of HAND2–AS1 [110]. In contrast, the expression of miR–4516 is shown to be downregulated in CAFs compared with normal fibroblasts and CAF–derived EVs often promote the proliferation of TNBC cells via upregulating the expression of the miR–4516 target FOSL1 [111]. Circular RNA HIF1A (circHIF1A) is enriched in the plasma EVs of TNBC patients and was shown to promote tumor growth via the activation of the PI3K/AKT pathway and downregulation of p21 [112]. Figure 4 illustrates how EVs play their part in the growth of TNBC.

### 4.2. EVs in TNBC Metastasis

EVs also participate in the metastatic dissemination of human TNBC. For example, in response to chemotherapy, TNBC cells release a significant number of PS + EVs, which trigger an increase in vascular permeability, leading to the transendothelial migration (TEM) of cancer cells [113]. Moreover, the serum EVs of TNBC patients are enriched with tetraspanins, including CD151, which was shown to stimulate the migration and invasion of TNBC cells [114]. Again, circulating EVs of TNBC patients are observed to be abundant with the cancer testis antigen SPANXB1, which downregulates the SH3GL2 expression in TNBC cells, thereby upregulating the expression of Rac1, FAK, and α–actinin, leading to TNBC metastasis [115]. Furthermore, the level of miR–9 and miR–155 is well–elevated in the TNBC cells MDA–MB–231–derived EVs, which target PTEN and DUSP14 in less metastatic cells, namely, MCF–7, leading to the enhancement of MCF–7 metastatic potential [116]. In addition, EVs from TNBC cells often show an upregulation of miR–939, which targets VE–cadherin in endothelial cells, thereby triggering increased vascular leakage to promote transendothelial migration of the tumor [117]. Serum EVs of TNBC patients are overexpressed with circPSMA1, which acts as a “miRNA sponge” to absorb miR–637, thereby releasing the inhibitory effect of miR–637 on AKT1 to stimulate β–catenin–driven expression of cyclin D1, ultimately leading to the enhanced proliferation, migration, and metastasis of TNBC cells [118]. Figure 4 also delineates how EVs contribute to the metastasis of human TNBC.

## 5. Blood Coagulation Is Associated with TNBC Progression

Blood coagulation and cancer are intrinsically related; enhanced thrombosis is often observed in various types of cancer, which contributes to the morbidity and mortality of cancer patients [119]. On the other hand, our studies, for the first time, found that coagulation proteases trigger the progression of human TNBC via the secretion of EVs. We demonstrated that coagulation protease factor VIIa (FVIIa) binds to its primary receptors, namely, TF and TF–FVIIa, via the activation of protease–activated receptor 2 (PAR2) [120], which stimulates the release of EVs from the human TNBC cells MDA–MB–231 [18]. The study indicates that TF–FVIIa–PAR2–mediated generation of EVs is dependent on three independent signaling pathways: the PI3K/AKT/p38–MAPK/MK2/HSP27 pathway, MEK^1/^_2_/ERK^1^/_2_/MLCK/MLC2 pathway, and ROCKII pathway [18]. The crosstalk of these three pathways promotes actomyosin reorganization, which critically regulates the release of EVs from the TNBC cells [18]. The study also revealed that TNBC–EVs are readily taken up by the non–metastatic breast cancer cells MCF–7, thereby triggering the induction of metastatic potential to EVs–fused recipient MCF–7 cells [18]. In a separate study, we investigated that another protease, namely, trypsin, which is well–known for the activation of PAR2, also promotes the generation of EVs from human TNBC cells via AKT/Rab5–dependent actin rearrangements and these EVs induce metastasis to MCF–7 cells [17]. In a continuation study, we revealed that FVIIa–EVs from TNBC cells are enriched with miR–221, which targets PTEN in the recipient MCF–7 cells, leading to the activation of AKT/NF–ĸB pathway to promote the expression of the EMT–associated transcription factors snail and slug [11]. EMT–transcription factors, in turn, trigger the induction of the mesenchymal markers N–cadherin and vimentin, while prohibiting the expression of the epithelial marker E–cadherin, ultimately leading to EMT [11]. The induction of EMT not only enhances the proliferation, migration, and invasion of EVs–fused recipient MCF–7 but also confers resistance of MCF–7 against the chemotherapeutic apoptosis–inducing drug cisplatin [11]. The incorporation of an anti–miR–221 inhibitor into EVs not only reverses the EV–triggered induction of EMT in MCF–7 but also downregulates EV–mediated MCF–7 proliferation, migration, invasion, and drug resistance [11]. Consistent with our in vitro observations, our in vivo data also show that FVIIa–EVs from TNBC cells promote the growth and metastasis of MCF–7 tumors, and the introduction of anti–miR–221 significantly attenuates EV–induced tumor growth and metastatic dissemination [11]. These studies were further strengthened by our human patients’ data, which indicate that compared with normal healthy controls, EVs from TNBC patients’ plasma also stimulate the miR–221–dependent growth and metastasis of non–metastatic cells in both in vitro and in vivo settings [11]. Figure 5 summarizes how EVs from FVIIa–treated TNBC cells promote the growth and metastasis of non–metastatic breast cancer cells, depending on the miR–221 transfer.

## 6. Role of EVs as a Biomarker for TNBC

As EVs often bear the cargo molecules of the cells from which they arise, they often serve as effective biomarkers in various pathophysiological conditions, including TNBC. The present section briefly summarizes the role of EVs as a biomarker in different types of TNBC (Table 4). Figure 6 also briefly delineates how EVs serve as a biomarker for TNBC in different ways.

### 6.1. EVs’ Proteins as a TNBC Biomarker

EVs from the TNBC cells MDA–MB–231 were shown to be enriched with extracellular growth factor receptor (EGFR) ligands, such as amphiregulin (AREG), which promote the invasiveness of recipient breast cancer cells, thereby being considered a biomarker for TNBC [121]. The expression of the deubiquitinase UCHL_1_ is significantly higher in EVs derived from TNBC cells and plasma, which stimulates the migration and extravasation of the breast cancer cells by upregulating TGFβ signaling via protecting both TGFβ type I receptors and SMAD2 from ubiquitination [122]. Moreover, the chemotherapeutic drug paclitaxel (PTX) induces the release of Survivin + EVs from MDA–MB–231 cells, which promote the growth and survivability of PTX–exposed fibroblasts and other breast cancer cells, which essentially serve as a TNBC biomarker [123]. In addition, serum EVs of TNBC patients are enriched with CD151, which was shown to induce the migration and invasion of TNBC cells and is thereby considered a predictive biomarker for TNBC [114].

### 6.2. EVs’ miRNAs as a Biomarker for TNBC

miRNAs are small (~22 nt long) non–coding RNA molecules that play a major role in tumor progression including TNBC [128]. Moreover, these miRNAs, which are carried by the EVs, serve as a prognostic and diagnostic biomarker for TNBC. For example, cisplatin (DDP)–resistant MDA–MB–231–cell–derived EVs transfer miR–423–5p to other breast cancer cells, leading to the induction of P–glycoprotein (P–gp), enhancement of migration and invasion, and downregulation of apoptosis [124]. Moreover, TNBC–associated macrophages release miR–223 + EVs, which promote the invasion of breast cancer via the Mef2c–β–catenin pathway, and thus, miR–223–loaded EVs are considered a prognostic and diagnostic biomarker for TNBC [125]. In addition, TNBC–EVs often show a higher expression of miR–10b, which induces the invasiveness of non–malignant cells by targeting HOXD10 and KLF4, and miR–10b–enriched TNBC–EVs often serve as a predictive biomarker for TNBC [126]. Furthermore, the EV–mediated transfer of miR–105 from the TNBC cells MDA–MB–231 to endothelial cells induces endothelial barrier permeability via targeting the expression of the tight junction protein ZO–1, thereby facilitating TNBC metastasis [127]. Hence, miR–105–containing TNBC–EVs are often considered a biomarker for early–stage TNBC.

## 7. Emerging Role of EVs in TNBC Immune Regulation

EVs from TNBC cells are often shown to actively participate in the immune regulation of TNBC. However, EVs released from different cells of the TME, such as stromal cells and activated immune cells, also contribute to TNBC immune regulation. Table 5 briefly summarizes EVs’ contribution to immune regulation in the context of TNBC. For example, TNBC–EVs are enriched with CCL5, which is readily transferred to naive macrophages, leading to their transformation into tumor EV–educated macrophages (TEMs) [129]. TEMs, in turn, release various factors, including IFNγ, CXCL1, OPN, CTLA–4, TGFβ, HGF, and CCL19, thereby inducing stromal remodeling and immune infiltration to facilitate TNBC metastasis [129]. Moreover, TNBC–EVs are also reported to induce macrophage polarization into type 2 (M2) phenotypes, thereby creating a favorable environment for the lymph node (LN) metastasis of TNBC [130]. Again, activated T–cells that express programmed death 1 (PD–1) often release PD–1+ EVs that interact with programmed death–ligand 1 (PD–L1)–containing TNBC cells or TNBC–EVs, thereby preventing direct interaction between T–cells/TNBC cells via PD–1/PD–L1, ultimately attenuating the PD–L1–triggered suppression of activated T–cells by TNBC cells [131]. On the other hand, oscillatory strain (OS) enhances the release of TNBC–EVs that are positive for PD–L1, which not only promotes the enrichment of myeloid–derived suppressor cells (MDSCs) and M2 macrophages in the TME but also reduces the abundance of CD8+ T–cells [132]. Figure 7 depicts how EVs are associated with the immune regulation of TNBC.

## 8. Emerging Role of EVs in TNBC Immunotherapy

Immunotherapy is the major segment of cancer therapies that involves the immune cells of the body fighting against the cancer cells [133]. The advantages of cancer immunotherapy over other therapeutic approaches include fewer side effects, long–term protection, targeting each mutation in the cells, and being potentially effective against every type of cancer [134]. Although immunotherapy has made significant promises against different cancer types, its role in the treatment of human TNBC is still in the initial stages. However, a few successful implementations of immunotherapeutic approaches against human NBC are ongoing.

### 8.1. Immunotherapeutic Approaches against Human TNBC

Although chimeric antigen receptor T–cell (CAR–T) therapy has become more promising against different types of hematological tumors, its application against TNBC has been far more challenging [135]. On numerous occasions, the PD–1/PD–L1 mechanism was shown to be an effective immunotherapeutic target against TNBC. Pembrolizumab, which is the FDA–approved immune checkpoint inhibitor, and monoclonal antibody, which targets PD–1, were in a phase I clinical trial against TNBC [100]. Atezolizumab, which is an anti–PD–L1 antibody, was also shown to be tolerated with durable clinical benefits against metastatic TNBC [136]. However, in the phase III clinical trial IMpassion131, atezolizumab did not pass through due to its limited clinical benefits [137]. Another FDA–approved immunotherapeutic agent, namely, sacituzumab govitecan (IMMU–132), which is a tumor–associated calcium signal transducer 2 (TROP2) antibody, in conjugation with SN–38, which is the topoisomerase/inhibitor chemotherapy, showed promising effects in phase I and II clinical trials against metastatic TNBC and is now in clinical trial III [138,139]. At present, thirteen immunotherapy–based approaches are registered against TNBC in clinical trials, and among them, ten are in phase I. In addition, fifty–five clinical trials have been registered against TNBC that involve different chemotherapeutic regimens, in combination with immunotherapy [140].

### 8.2. EVs in TNBC Immunotherapy

Cytotoxic immune cells like natural killer (NK) cells and CD8+ cytotoxic T–lymphocytes (CTLs) have an amazing ability to find and kill cancer cells [141]. This has been used to make immunotherapies for TNBC [142]. However, due to their low penetrance into the tumor to reach the cancer cells, as well as their restricted efficacy, these mechanisms have limited therapeutic applications [142,143]. The advent of EVs, which can penetrate even solid tumors, has led researchers to employ EVs derived from NK–cells or CTLs in the treatment of different TNBC types. TNBC tumors often express PD–L1 on the cell surface, whose receptor PD–1 is expressed on tumor–infiltrating lymphocytes (TILs). The interaction of tumor cells with TILs via PD–L1/PD–1 not only attenuates TIL proliferation but also leads to the apoptosis of TILs, thereby contributing to the immune evasion mechanisms of the TNBC [142]. Qiu et al. demonstrated that PD–1+ EVs are released from TILs, which interact with either the cell surface or EVs’ PD–L1, thereby preventing the TILs–TNBC cells interaction through PD–L1/PD–1, ultimately leading to the attenuation of the PD–L1–mediated suppression of TILs activity [131]. This mechanism opens a new immunotherapeutic strategy to enhance TIL activity, thereby facilitating the killing of TNBC cells (Figure 7).

Unlike CTLs, NK–cells, independent of major histocompatibility complex I (MHC–I) recognition, exert cytolytic activity via the downregulation of tumor growth and metastasis [143]. Similar to NK–cells, NK–EVs were also shown to be positive for killer proteins like FasL, perforin, and granzyme [144], and elicit cytotoxicity against solid tumors [145], via the transfer of killer proteins (Figure 8A) [146]. Although, so far, NK–EVs have been shown to be effective against various solid tumors, their effects still need to be validated in the context of TNBC.

On several occasions, EVs carrying bioactive molecules, such as miRNAs, were shown to be effective therapeutic targets in the treatment of TNBC [147]. For example, let–7, encapsulated within aptamer–AS1411–modified EVs, is readily taken up by nucleolin–positive TNBC cells, leading to a significant reduction in tumor growth [147]. Again, bone marrow stromal EVs enriched with signature miRNA molecules (miR–222, miR–197, miR–127, and miR–223) often reduce the proliferation of TNBC cells [148]. Furthermore, adipose–tissue–derived MSCs promote the apoptosis of TNBC cells via the transfer of miR–424–5p–enriched EVs [149]. In addition, MSC–EVs also bear miR–381, which decreases the viability, migration, and invasion of TNBC cells via the downregulation of the Wnt pathway, expression of Twist and Snail, and subsequent inhibition of EMT [150]. Figure 8B summarizes how EVs’ miRNAs serve in the therapeutic implications of human TNBC.

In addition to the above, bioengineered EVs from the TNBC cells themselves often turn out to be more effective against TNBC [151]. For example, TNBC–exosome–encapsulated doxorubicin was shown to be more effective against TNBC than its free form, with reduced cardiac toxicity [151]. On the other hand, miR–9 and miR–155–loaded EVs from TNBC cells often transfer their metastatic phenotype to the non–metastatic cells MCF7 via the downregulation of the respective targets PTEN and DSUP14 [116]. Furthermore, regarding TNBC cells, HCC1806–derived EVs not only promote the proliferation but also confer resistance in the normal breast epithelial cells MCF10A against the therapeutic drugs docetaxel and doxorubicin [107]. A recent study indicated that EVs released from TNBC cells that were bioengineered to express a high–affinity variant of the human PD–1 protein (havPD–1) and knockdown intrinsic PD–L1 and β2–microglobulin reduce tumor proliferation and promote apoptosis via the downregulation of PD–L1–dependent T–cell suppression [152]. The addition of a PARP inhibitor with PD–1 + EVs further improves the treatment efficacy [152]. Moreover, EVs from HEK293F cells are also shown to target TNBC tumors when bioengineered to contain verrucarin A (Ver–A) followed by surface tagging with anti–EGFR/CD47 mAbs [153].

## 9. EVs in Clinical Trials: The Translational Significance in Cancer

Due to significant success in preclinical studies indicating the enormous diagnostic potential of EVs, EVs have been registered in several clinical trials in the context of different cancers, as discussed in Table 6 (www.clinicaltrials.gov, accessed on 18 September 2023).

## 10. Conclusions, Future Direction and Challenges

TNBC is considered the most aggressive form of human breast cancer and accounts for ~10–20% of all breast cancer phenotypes. The paucity of PR and HER2 on the TNBC cells renders conventional therapeutic regimens ineffective against human TNBC. A growing body of evidence indicates that EVs play a pivotal role in the treatment of various pathophysiological conditions due to their extraordinary capability to deliver bioactive molecules to a wide variety of cells. Additionally, bioengineered EVs show potential in the fight against different types of human cancer. However, the functional application of EVs in the treatment of human TNBC is still in its initial stages. The present review highlighted the in–depth understanding of the crucial roles of EVs in the growth and metastatic dissemination of human TNBC. Moreover, how the EVs contribute as biomarkers for the early detection and progression of human TNBC is also illustrated. In addition to these, the immune regulation of EVs in the context of human TNBC is emerging and was shown to be highly effective. EV–based immunotherapeutic interventions also delineate beneficial effects in combating human TNBC. Despite these EV–associated immunotherapeutic advancements, the mechanism and function of the EVs in TNBC tumorigenesis are yet to be completely explored. A thorough and significant study needs to be performed to understand the diagnostic, prognostic, and therapeutic values of EVs in clinical settings. Moreover, EV–based combinational therapy alongside immunotherapy could be highly effective against human TNBC.

Despite the abovementioned promises, EV–based therapy against TNBC still faces significant challenges. The conventional protocols for the isolation and characterization of EVs often result in EV yields with high variability. Moreover, contamination by cellular fragments, impurities, and similar–sized particles further add to the variability. Again, the role of EVs in the epigenetic regulation of the TNBC needs to be further investigated. Additionally, in the TME, several cells communicate with the tumor cells and vice versa via the release of EVs; hence, it is very difficult to understand the crucial roles of the EVs in a particular cell type under a given physiologic condition. However, despite these shortcomings, EVs are still considered a promising biomarker and therapeutic means against TNBC [167].

In addition to the above, the treatment of human TNBC itself has a few significant limitations. The major metabolic pathways involved in the progression of TNBC that could be targeted converge with one or a few HUB proteins and bottleneck proteins that include PTEN, AKT1, MAPK1, MAPK3, EGFR, TP53, UBC, HRAS, RPS27A, GRB2, UBA52, and SRC. These proteins also play pivotal roles in the normal physiological functions of almost every cell; hence, targeting these proteins often interferes with the normal physiological functions of the system. Therefore, at present, the identification of specific molecules is a prerequisite and of prime importance in targeting TNBC tumors without affecting normal physiology. Without knowledge of specific molecular mechanisms, designing the drugs that combat TNBC becomes almost impossible. Moreover, several post–translational modifications (PTMs) affecting protein functions and activity further add to the complexity. The heterogeneity of TNBC can also be influenced by PTMs, which again complicates the matter for the development of specific targeted therapies, including emerging immunotherapies. The lack of a comprehensive and dynamic view of protein interactions, PTMs, and metabolic functions not only impedes understanding the complex behavior of human TNBC but also restricts the development of specific therapeutic targets. However, the emerging development of alternating approaches and technologies, as well as the advancement of genomics, proteomics, metabolomics, and other disciplines, certainly hold promise for a better understanding of TNBC complexity, thereby developing potential therapeutic targets.

## Figures and Tables

**Figure 1 cancers-15-04879-f001:**
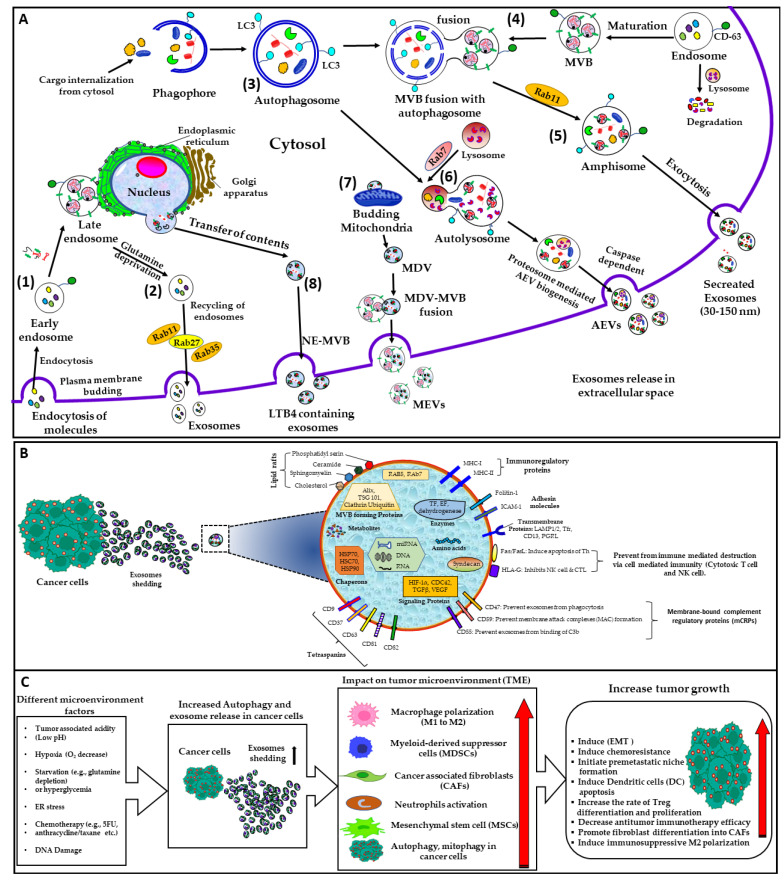
The biogenesis of different multivesicular bodies (MVBs) and composition of EVs and their immunoregulatory roles in tumor microenvironment. (**A**). Biogenesis of various types of multivesicular bodies (MVBs). (1) The canonical MVB biogenesis starts at the plasma membrane, where cargos are sorted efficiently in an early endosome via endocytosis, which further matures into the multivesicular appearance of late endosomes (MVBs). (2) The recycling of endosomes during glutamine deprivation. (3) After sequestering a portion of the cytosol called a phagophore, it develops further into CD63–positive and LC3–positive MVBs called autophagosomes that fuse with multivesicular bodies (MVBs) that originate from late endosome through endocytosis of plasma membrane and form amphisomes (4,5) that release into extracellular space via exocytosis. (6) Autophagosomes fuse with lysosomes to form autolysosomes and release AEVs into the extracellular space in a caspase–3–dependent manner. (7) The budding of mitochondria into mitochondria–derived vesicles (MDVs) and their fusion with MVBs, followed by the release of mitochondria–derived vesicle (MDV) exosomes containing mitochondrial proteins and mitochondrial DNA (mtDNA). (8) Nuclear–envelope–derived MVBs (NEMVBs) are produced via the outward budding of the inner nuclear envelope, which is followed by the release of exosomes that contain leukotriene B4 (LTB4). (**B**) General structure of exosomes with the surface markers of an exosome. Exosomes originate from MVBs and carry different noncoding RNA, including miRNA, lncRNA, and circRNA; proteins (such as HSP 60, 70, and 90); and so on. Additionally, exosomes also carry different molecules that provide them immune tolerance (i.e., help them avoid being destroyed by the immune system and complement), such as CD47, CD55, and CD59, which prevent exosomes from being destroyed by the complement cascade. HLA–G and Fas/FasL molecules also protect exosomes from being destroyed by cytotoxic T and NK cells. These contents have a significant impact on tumor development. (**C**) In various microenvironments (e.g., low pH, hypoxic conditions, starvation, and ER stress), cancer cells release exosomes and may induce autophagy in recipient cells, which can further induce cell proliferation, EMT, and chemoresistance due to a significant impact on different components of the tumor microenvironment, like neutrophils activation, macrophage polarization, fibroblast differentiation into CAFs, and promote premetastatic niche formation. Red arrow indicate increased (upward) level.

**Figure 2 cancers-15-04879-f002:**
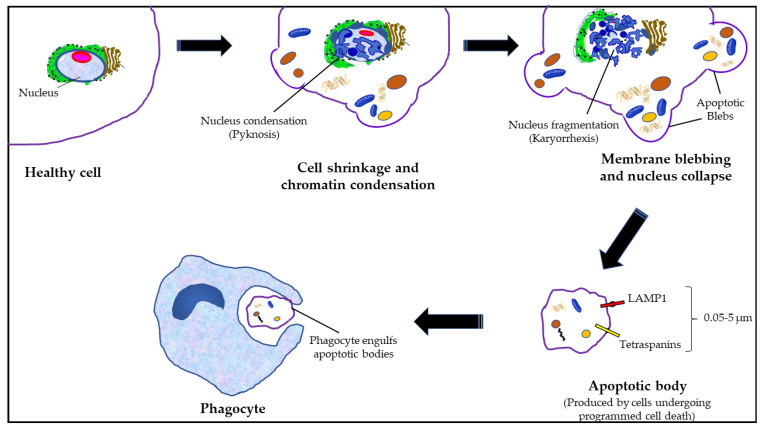
An apoptotic bodies (between 0.05–5 µm in diameter), which is produced by dying cells during the later stages of apoptosis, is released into the extracellular space, and eliminated by neighboring cells.

**Figure 3 cancers-15-04879-f003:**
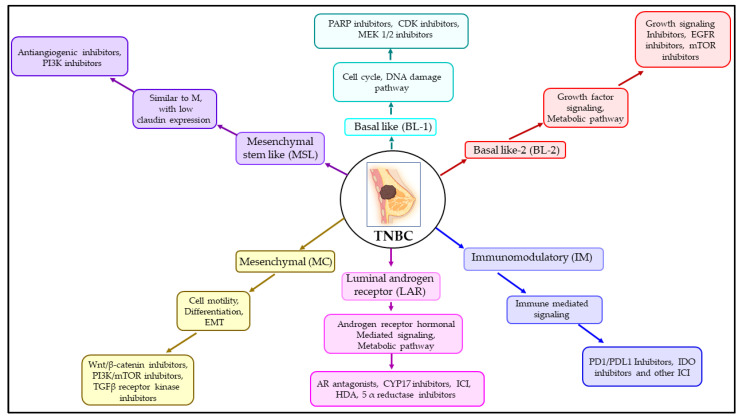
TNBC subtypes, their general characteristics, and treatment measures. Based on gene expression profiles, TNBC is categorized into six subtypes, i.e., BL–1, BL–2, IM, LAR, M, and MSL. Higher expressions of genes related to the cell cycle and the DNA damage pathway are linked to BL–1. PARP inhibitors and platinum compounds are used to treat BL–1 (dark green arrow). BL–2 is accompanied by higher expressions of growth–factor–signaling and metabolic–pathway–associated genes and can be treated with growth–signaling inhibitors (dark red arrow). IM is related to elevated expression of genes associated with immune cellular processes; the treatment measures include immune checkpoint inhibitors (blue arrow). LAR, on the other hand, is associated with higher relative expression of hormonal regulation and metabolic pathway genes and can be targeted by androgen receptor agonists (purple arrow). Elevated expressions of cell motility, differentiation, and EMT genes are observed in MC, which can be treated with Wnt/β–catenin pathway, PI3K/mTOR pathway, and TGFβ receptor kinase inhibitors (dark yellow arrow). MSL is very similar to M; however, claudin expression was shown to be significantly lower in MSL compared with M (violet arrow).

**Figure 4 cancers-15-04879-f004:**
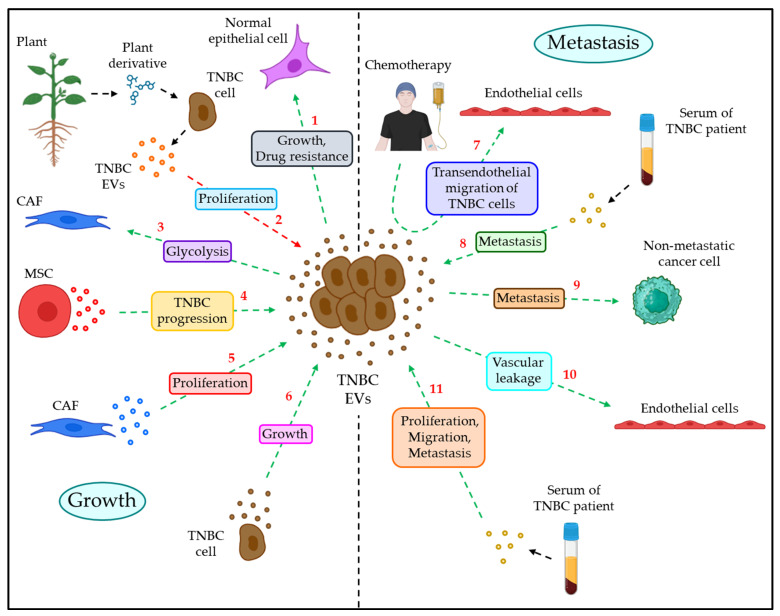
The role of EVs in the growth and metastasis of human TNBC. Growth: (1) TNBC–EVs induce growth and drug resistance in normal epithelial cells. (2) Plant derivative–induced TNBC cell–derived EVs inhibit TNBC proliferation. (3) TNBC–EVs also enhance glycolysis in CAFs. (4) MSC–EVs promote TNBC progression. (5) CAF–EVs also induce TNBC proliferation. (6) TNBC–EVs often trigger the growth of TNBC in an autocrine manner. Metastasis: (7) In response to chemotherapy, TNBC cells release EVs, which act on endothelial cells to promote transendothelial migration of TNBC cells. (8) Serum–EVs of TNBC patients promote TNBC metastasis. (9) TNBC–EVs also promote metastasis of non–metastatic breast cancer cells. (10) TNBC–EVs also enhance vascular permeability upon acting on endothelial cells. (11) Serum–EVs of TNBC patients again trigger proliferation, migration, and metastasis of TNBC cells. The green dotted arrows indicate phenomena associated with TNBC progression, whereas red dotted arrow denotes prevention of TNBC progression.

**Figure 5 cancers-15-04879-f005:**
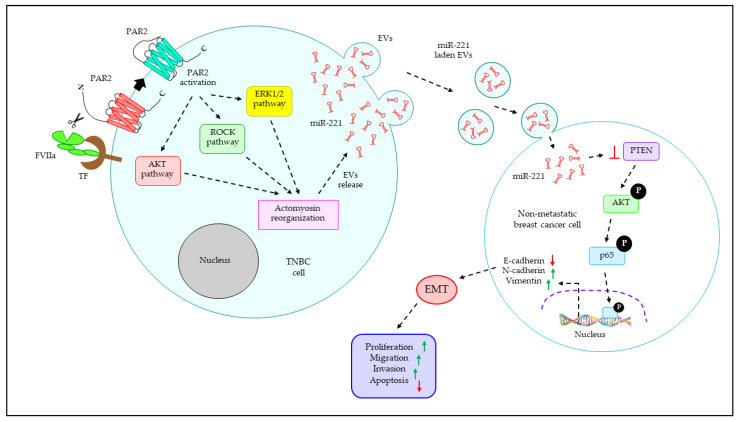
TF–FVIIa–driven PAR2 activation triggers EV generation from human TNBC cells, which promotes EMT in non–metastatic breast cancer cells, leading to the induction of proliferation, migration, invasion, and anti–apoptosis. TF–FVIIa–mediated cleavage of PAR2 in human TNBC cells leads to the intracellular activation of AKT, ROCK, and ERK1/2 pathways independently, which promotes the release of EVs packaged with miR–221 via actomyosin reorganization. TNBC–EVs are taken up by non–metastatic breast cancer cells, thereby delivering miR–221 into the recipient cells. In recipient cells, miR–221 targets PTEN, leading to the activation of AKT/p65–signaling pathway in which the activated (via phosphorylation) p65 enters the nucleus to stimulate the upregulation of N–cadherin and vimentin while downregulating the expression of E–cadherin. As a result of this, EMT is induced, which ultimately leads to the promotion of proliferation, migration, invasion, and anti–apoptosis of EV–fused recipient cells. Red arrows indicate decreased and green arrow indicate increased levels.

**Figure 6 cancers-15-04879-f006:**
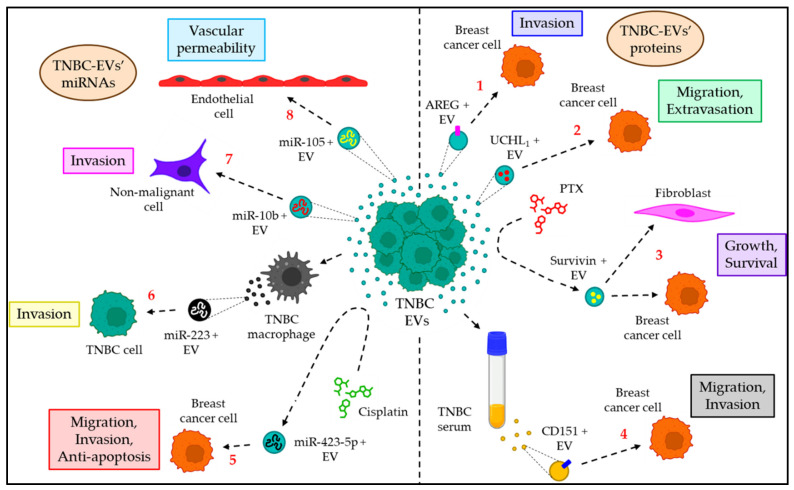
The role of EVs as a biomarker for TNBC. EV–carried proteins often serve as TNBC biomarkers. (1) AREF + TNBC–EVs promote breast cancer invasiveness. (2) UCHL_1_ + TNBC–EVs trigger migration and extravasation of breast cancer cells. (3) PTX–treated TNBC cells release Survivin + EVs, which promote the growth and survival of fibroblasts and breast cancer cells. (4) TNBC serum is enriched with CD151 + EVs, which induce migration and invasion of breast cancer. EV–transported miRNAs also contribute as a biomarker for TNBC. (5) Cisplatin–resistant TNBC–EVs carry miR–423–5p, which promotes migration, invasion, and anti–apoptosis of breast cancer cells. (6) TNBC–associated macrophage–derived EVs are positive for miR–223, which stimulates invasion of TNBC cells. (7) TNBC–EVs enriched with miR–10b also promote invasion of non–malignant cells. (8) miR–105–loaded TNBC–EVs increase endothelial permeability, thereby contributing to TNBC metastasis.

**Figure 7 cancers-15-04879-f007:**
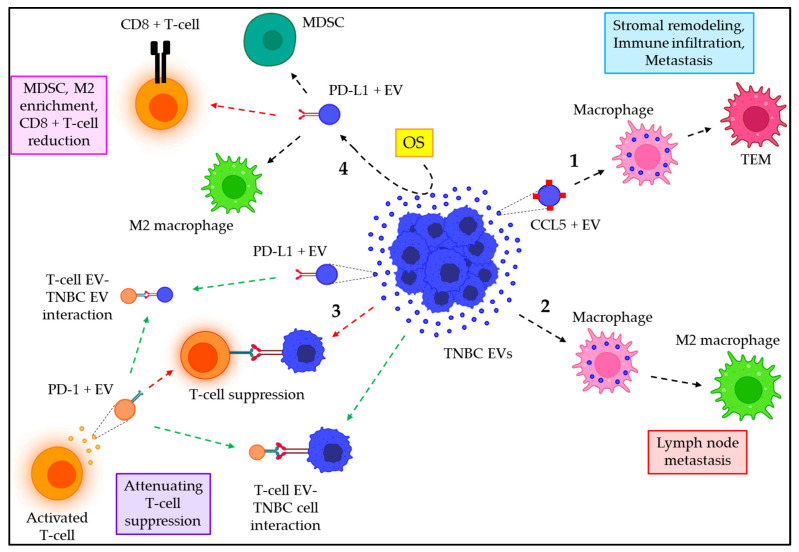
The role of EVs in TNBC immune regulation. (1) TNBC–EVs are positive for CCL5, which triggers the conversion of macrophages into TEMs, leading to stromal remodeling and immune infiltration, ultimately contributing to TNBC metastasis. (2) TNBC–EVs promote M2 polarization of macrophages, which leads to lymph node metastasis of TNBC. (3) Activated T–cells release PD–1+ EVs that bind to PD–L1, either on TNBC cells or TNBC–EVs (defined as green dotted arrows), thereby preventing the PD–1/PD–L1–dependent interaction of T–cells and TNBC cells (shown as red dotted arrows), because of which TNBC–mediated T–cell suppression is reduced. (4) OS triggers the release of PD–L1+ EVs from TNBC cells, which not only enrich MDSCs and M2 macrophages in the TME but also reduce the population of CD8+ T–cells.

**Figure 8 cancers-15-04879-f008:**
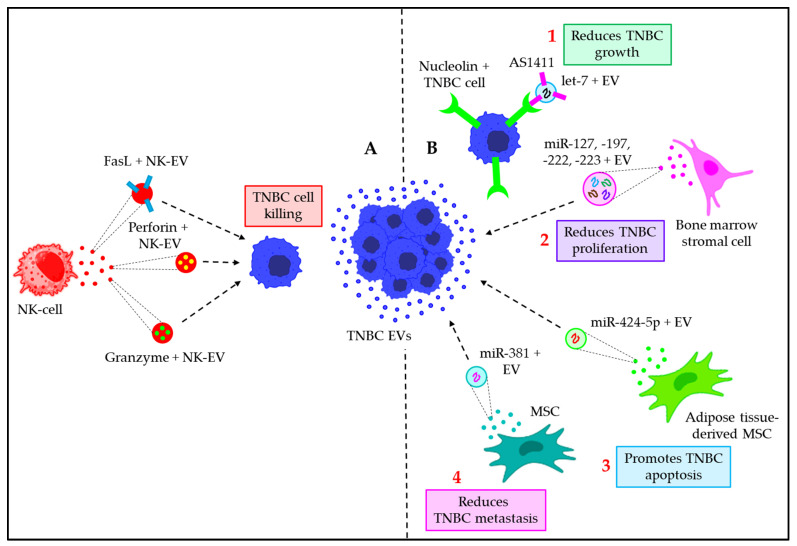
The role of EVs in TNBC immunotherapy. (**A**) NK–EVs in TNBC immunotherapy. EVs are shown to be enriched with FasL, Perforin, and Granzyme, which trigger the killing of TNBCs. (**B**) EVs’ miRNAs in TNBC immunotherapy. (1) let–7–containing EVs with aptamer (AS1411) modification target nucleolin + TNBC cells, resulting in the downregulation of TNBC growth. (2) Bone marrow stromal cell–derived EVs are abundant with miR–127, –197, –222, and –223, which reduce TNBC proliferation. (3) Adipose–tissue–derived MSCs release miR–424–5p–bearing EVs, which promote TNBC apoptosis. (4) MSC–EVs are rich in miR–381, which reduces the metastatic potential of TNBC.

**Table 1 cancers-15-04879-t001:** General characteristics of different types of EVs.

Type of EVs	Marker/s	Size (Diameter)	Biogenetic Mechanism	Reference
Microparticles (MPs)	Tetraspanins, CD41, CD63	100 nm–1 µm	Evagination of the plasma membrane followed by pinching to release the MPs outside the cells.	[53,54]
Exosomes	Alix, TSG101, HSP90β, HSC70, CD63, Syntenin 1	30–150 nm	Conventional mechanism of exosome biogenesis includes the invagination of the early endosomal membrane to form exosomes, which fuse with the plasma membrane to release the exosomes from the cells.	[55,56,57]
Apoptotic bodies	HSP60, GRP78, Annexin V (phosphatidylserine)	50 nm–5 µm	Cellular contraction is induced during apoptosis, which produces a hydrodynamic force that triggers the separation of the plasma membrane from the cytoskeleton, leading to the release of apoptotic bodies from the cells.	[55,58,59]

*Abbreviations:* TSG101, tumor susceptibility gene 101; Alix, ALG–2–interacting protein X; HSP, heat shock protein; HSC70, heat shock cognate protein 70; CD, cluster of differentiation; GRP78, glucose–regulated protein 78.

**Table 2 cancers-15-04879-t002:** Subtyping of breast cancer, based on immunohistochemistry analysis.

Subtype	Immunohistochemistry Profile	Other Features	References
Luminal A	ER+, PR+, HER2−	Ki–67−	[76]
Luminal B	ER+, PR+, HER2+	Ki–67+	[76]
HER2 high	ER−, PR−, HER2+	Ki–67+	[81]
Normal–like	ER+, PR+, HER2−	Ki–67−	[81]
Basal–like (TNBC)	ER−, PR−, HER2−, basal markers	Ki–67+, (EGFR, CK5/6)	[76]
Claudin low (TNBC)	ER−, PR−, HER2−	Low Ki–67, E–cadherin, Claudin 3, 4, 7	[93]

*Abbreviations:* ER, estrogen receptor; PR, progesterone receptor; HER2, human epidermal growth factor receptor–2; TNBC, triple–negative breast cancer; EGFR, extracellular growth factor receptor; CK, cytokeratin; −, negative; +, positive.

**Table 3 cancers-15-04879-t003:** The role of EVs in the growth and metastasis of human TNBC.

TNBC Property	Mechanism of Action	References
Growth	HCC1806–EVs promote the growth and drug resistance to MCF10A cells via the involvement of PI3K/AKT, MAPK, and HIF1A pathways.	[107]
	DET– and DETD–35–treated MDA–MB–231–derived EVs inhibit the proliferation of MDA–MB–231 cells by downregulating cell adhesion, migration, and angiogenesis.	[108]
	ITGB4 + EVs from MDA–MB–231 cells trigger the glycolytic pathway of CAFs via BNI3PL–dependent mitophagy and lactate production, which, in turn, promote tumor growth.	[109]
	MSC–EVs transfer miR–106a–5p to the TNBC tumor and induce tumor progression via downregulating HAND2–AS1.	[110]
	CAF–EVs show a relatively lower expression of miR–4516, which promotes the proliferation of TNBC cells by upregulating the miR–4516 target, namely, FOSL1 expression.	[111]
	CircHIF1A + EVs from human TNBC promote tumor growth via the upregulation of the PI3K/AKT pathway while downregulating p21.	[112]
Metastasis	In response to chemotherapy, TNBC cells release a significant number of PS + EVs, which induce endothelial barrier permeability, hence helping with the transendothelial migration of cancer cells, contributing to TNBC metastasis.	[113]
	Serum EVs of TNBC patients are enriched with CD151, which triggers the migration and invasion of TNBC cells.	[114]
	Circulating EVs of TNBC patients are enriched with SPANXB1, which downregulates SH3GL2 expression in TNBC cells, and upregulates Rac1, FAK, and α–actinin expression, leading to TNBC metastasis.	[115]
	MDA–MB–231–EVs are abundant with miR–9 and miR–155, which target PTEN and DUSP14 in MCF–7, thereby inducing the metastatic potential of recipient MCF–7 cells.	[116]
	TNBC–EVs are enriched with miR–393, which targets VE–cadherin in endothelial cells, leading to enhanced vascular leakage, and thereby facilitating transendothelial migration of tumor cells.	[117]
	Serum EVs of TNBC patients are enriched with circPSMA1, which absorbs miR–637 to release the inhibitory function on AKT1 and leads to the downstream activation of β–catenin and cyclin D1 to trigger the proliferation, migration, and metastasis of TNBC cells.	[118]

*Abbreviations:* PI3K, phosphoinositide 3–kinase; MAPK, mitogen–activated protein kinase; HIF1A, hypoxia–inducible factor 1α; DET, deoxyelephantopin; DETD–35, DET derivative 35; ITGB4, integrin β4; CAF, cancer–associated fibroblast; BNI3PL, B–cell leukemia/lymphoma 2 protein (Bcl–2) interacting protein 3 like; MSC, mesenchymal stem cell; HAND2–AS1; heart and neural crest derivatives expressed 2 antisense RNA 1; FOSL1, Fos like 1; PS, phosphatidylserine; CD, cluster of differentiation; SPANXB1, sperm protein associated with the nucleus on the X chromosome (SPANX) family member B1; SH3GL2, SH3 domain containing GRB2 like 2; Rac1, Ras–related C3 botulinum toxin substrate 1; FAK, focal adhesion kinase; PTEN, phosphatase and tensin homolog; DUSP14; dual specificity phosphatase 14; VE–cadherin, vascular endothelial cadherin; PSMA1, proteasome 20S subunit α1.

**Table 4 cancers-15-04879-t004:** The role of EVs as a biomarker for human TNBC.

EVs’ Cargo Type	Name of the Cargo	Origin	Function	References
Protein	EGFR ligands	MDA–MB–231	EVs’ EGFR ligands, such as AREG, promote the invasiveness of recipient breast cancer cells.	[121]
	UCHL_1_	TNBC cells, TNBC plasma	EV–carried UCHL_1_ protects the TGFβ type I receptor and SMAD2 from ubiquitination, stimulating the migration and extravasation of the breast cancer cells.	[122]
	Survivin	MDA–MB–231	PTX–treated TNBC–derived EVs are enriched with Survivin, which promotes the growth and survivability of PTX–exposed fibroblasts and other breast cancer cells.	[123]
	CD151	TNBC serum	EVs from TNBC serum are enriched with CD151, which promotes the migration and invasion of TNBC cells.	[114]
miRNA	miR–423–5p	MDA–MB–231	EVs from cisplatin–resistant TNBC cells are positive for miR–423–5p, which induces P–gp expression, migration, invasion, and anti–apoptosis in other breast cancer cells.	[124]
	miR–223	MDA–MB–231	TNBC–associated macrophages release miR–223–containing EVs, which promote the invasion of breast cancer.	[125]
	miR–10b	TNBC cells	miR–10b + TNBC–EVs target HOXD10 and KLF4 in non–malignant cells to promote cell invasion.	[126]
	miR–105	MDA–MB–231	miR–105 + EVs target ZO–1 in endothelial cells, leading to increased vascular permeability and facilitating the metastasis of TNBC.	[127]

*Abbreviations:* EGFR, extracellular growth factor receptor; AREG, amphiregulin; UCHL_1_, ubiquitin C–terminal hydrolase 1; TGFβ, transforming growth factor beta; SMAD2, mothers against decapentaplegic homolog 2; PTX, paclitaxel; CD, cluster of differentiation; HOXD10, homeobox D10; KLF4, Krüppel–like factor 4; ZO–1, zonula occludens protein 1.

**Table 5 cancers-15-04879-t005:** The role of EVs in immune regulation of TNBC.

Donor Cells	Recipient Cells	EVs’ Cargo	Function	References
TNBC	Macrophages	CCL5	TNBC–EVs transport CCL5 to macrophages leading to the development of TEMs, which induce stromal remodeling and immune infiltration to facilitate TNBC metastasis.	[129]
TNBC	Macrophages		TNBC–EVs trigger macrophage polarization into M2 phenotypes, leading to LN metastasis.	[130]
Activated T–cells	TNBC cells or TNBC–EVs	PD–1	Activated T–cell–derived EVs release PD–1 + EVs, which bind to PD–L1 + TNBC cells or EVs, thereby preventing T–cell/TNBC cell interaction and attenuating immune suppression of T–cells by TNBC cells.	[131]
TNBC	MDSCs, M2– macrophages, CD8 + T–cells	PD–L1	OS stimulates the release of PD–L1 + EVs from TNBC cells, which enriches MDSCs and M2 macrophages in the TME and reduces CD8 + T–cells.	[132]

*Abbreviations:* CCL5, C–C motif chemokine ligand 5; M2 macrophage, type 2 macrophage; LN, lymph node; PD–1, programmed death 1; PD–L1, programmed death ligand 1; MDSC; myeloid–derived suppressor cell; OS, oscillatory strain; TME, tumor microenvironment; CD, cluster of differentiation.

**Table 6 cancers-15-04879-t006:** EVs in clinical trials in different types of cancer.

Trial Name	NCT Number	Characteristics	Cancer Types	EVs’ Source	Status	Outcome Measures	Refs.
Clinical Validation of an Urinary Exosome Gene Signature in Men Presenting for Suspicion of Prostate Cancer	NCT02702856	Duration: May 2014–June 2015 Population: 2000 men Age: >50	Prostate cancer	Urine	Completed	The ExoDx Prostate IntelliScore (EPI) EPI score was associated with low–risk pathology post–RP, with potential implications on informing AS decisions.	[154,155]
Diagnostic Accuracy of Circulating Tumor Cells (CTCs) And Onco–exosome Quantification in the Diagnosis of Pancreatic Cancer—PANC–CTC (PANC–CTC)	NCT03032913	Duration: February 2017–November 2017 Population: 20 with PDAC and 20 controls Age: 18 years and older	Pancreatic ductal adenocarcinoma	Circulation	Completed		[156]
microRNAs Role in Pre–eclampsia Diagnosis	NCT03562715	Duration: November 2016–December 2017 Population: 100 patients and 100 controls Females Age: 23 years to 35 years	Preeclampsia	Circulation	Completed	Liquid biopsy combining several biomarkers could provide a rapid, reliable, noninvasive decision–making tool in early, potentially curable pancreatic cancer.	[157,158]
Clinical Evaluation of the “ExoDx Prostate IntelliScore” (EPI)	NCT03031418	Duration: September 2016–September 2018 Population: 532 Patients Age: above 50 years of age Males	Prostate cancer	Urine	Completed	The expression of miRNAs 136, 494, and 495 in exosomes of peripheral blood and UCMSCs conditioned media.	[159]
Olmutinib Trial in T790M(+)NSCLC Patients Detected by Liquid Biopsy Using BALF Extracellular Vesicular DNA	NCT03228277	Duration: July 2017–July 2019 Population: 25 Males or females, aged at least 19 years.	NSCLC	BALF	Completed	ExoDx Prostate (IntelliScore) test can predict ≥ GG2 PCa at initial biopsy and defer unnecessary biopsies better than existing risk calculators and standard clinical data.	[160]
Pilot Study with the Aim to Quantify a Stress Protein in the Blood and in the Urine for the Monitoring and Early Diagnosis of Malignant Solid Tumors (EXODIAG)	NCT02662621	Duration: December 2015–April 2019 Population: 71 patients Age: 18 years and older	Cancer	Circulation	Completed	Assess the anti–tumor efficacy via objective response rate (ORR), disease control rate (DCR), and progression–free survival (PFS).	[161]
Pimo Study: Extracellular Vesicle–based Liquid Biopsy to Detect cancer Hypoxia in Tumors	NCT03262311	Duration: November 2017–September 2019 Population: 21 Age: ≥18 years	Hypoxia–induced cancer	Blood	Completed	HSP70 exosomes could be a powerful tool to diagnose cancer and guide.	[162]
The Sensitivity and Specificity of Using Salivary miRNAs in Detection of Malignant Transformation of Oral Lesions	NCT04913545	Duration: January 2020–August 2020 Population: 18 Age: 35 years to 70 years	Oral premalignant lesions	Saliva	Completed	Clinicians in therapeutic decision–making, improving patient care.	[163]
Clinical Evaluation of ExoDx Prostate (IntelliScore) in Men Presenting for Initial Prostate Biopsy	NCT04720599	Duration: June 2020–June 2021 Population: 120 males Age: 50 years and older	Urologic cancer	Urine	Completed	Not available.	
Serum Exosomal Long Noncoding RNAs as Potential Biomarkers for Lung Cancer Diagnosis	NCT03830619	Duration: Jan 2017–July 2021 Population: 1000 Age: 18 years to 75 years Location: Hubei, China	Lung cancer	Serum	Completed	Measuring the sensitivity and specificity of using the salivary miRNAs (412,512) to detect the malignant transformation in potentially malignant lesions.	[164]
Identification of New Diagnostic Protein Markers for Colorectal Cancer (EXOSCOL01)	NCT03895216	Duration: December 2018–December 2021 Population: 34 Age: 18 years and older	Bone metastasis	Plasma	Completed	EPI–CE provides information beyond standard clinical parameters and provides a better risk assessment prior to MRI of patients suspected of prostate cancer than the commonly used multiparametric risk calculators.	[165]
Exosomes Implication in PD1–PD–L1 Activation in OSAS (ExoSAS)	NCT03811600	Duration: March 2019–October 2020 Population: 90 Age: 18 years and older	Cancer, obstructive sleep apnea	Plasma	Completed	No study results are posted on ClinicalTrials.gov for this study.	[166]

*Abbreviations:* NSCLC, non–small–cell lung cancer; BALF, bronchoalveolar fluid; CRC, colorectal cancer.

## Data Availability

Not applicable.

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
