# Peer review of "Extracellular Vesicles in Triple–Negative Breast Cancer: Immune Regulation, Biomarkers, and Immunotherapeutic Potential"

_cancers, 2023, doi:10.3390/cancers15194879_

Round 1

Reviewer 1 Report

I have read the review and find it well done and organized but, if we exclude the tables, which are badly organized and incomprehensible to non-experts (and therefore need to be redone), there is something wrong. I'll be a bit redundant, but just to be understood.

When writing a review, it's not enough to describe the facts, you also need to give your expert opinion on today's limits and on the prospects for the future. Today, we always look for the molecular origin of cancers, but there are limits that must always be expressed. The limit lies because we still have a static view of metabolic functionality. For example, native proteins do not exist except sometimes. The real and functional proteins are the different proteoforms deriving from the native form with the function that the specific PTM modification assigns to it. The curators of many biomedical analysis platforms collapse any property of the proteoform on the native protein, which thus becomes a biological object with many supernumerary properties that it does not possess. This means that not only do we not recognize the proteoforms, but the spatio-temporal characteristics of each of them are missing. We do not know where, how, and when they act. All this makes it impossible to know the actual role of many proteins. The metabolic history that we build around a form of cancer, for these reasons, does not allow us, many times, to define the metabolic actors, but the history must be written anyway at all costs. Added to this is that metabolism is a complex network with a huge number of intersections that pass through the specific functional properties of two basic types of proteins, the HUB proteins and the bottleneck proteins.

These are proteins, on which the physical and functional interactions of dozens of different proteins converge, and which, carrying out their activity as exchangers, divert the output according to the metabolic context (as well as their own spatio-temporal characteristics). In triple breast cancer, some of them, such as PTEN, AKT1, MAPK1, MAPK3, TP53, EGFR, UBC, HRAS, RPS27A, GRB2, UBA52, SRC, are definitely all proteins with the role of HUB but the parameters topological systems, present in the many networks existing in the literature, also show them as protein bottlenecks, i.e., metabolic crossroads. Therefore, the current knowledge we have of metabolism, a highly dynamic system, is totally static and could contain inaccuracies. This is a limit to human knowledge to fight at the root diseases that are largely based on the microscopic alteration of deep molecular mechanisms that cannot be defined with macroscopic studies. Without the knowledge of the correct molecular mechanism, it is not possible to design drugs that fight this disease.

These considerations reflect a misunderstanding almost never made clear in the challenges facing the study of metabolism and molecular pathologies, related to the complexity of the metabolic function and the need to consider the specific proteoforms and their spatio-temporal properties, for an accurate understanding of these processes organic.

The static view of native proteins limits the understanding of their dynamic functionality and the role they play in biological processes. Post-translational modifications (PTMs) can affect protein function, and the metabolic context can vary their activity. All this creates virtual knowledge that pollutes actual knowledge, much more than predatory magazines because it goes unnoticed. Understanding the dynamics of proteoforms and interactions within the metabolic network is critical to gaining a comprehensive understanding of biological processes. Therefore, the complexity of the interactions within the metabolism between HUB and bottleneck proteins adds further challenges to understanding the system. These interactions can vary by context, and understanding how these proteins affect metabolic flux is critical to the development of targeted therapies.

Ultimately, the recognition of the challenges that the metabolic system presents is essential to guide research towards innovative solutions that can lead to a deeper understanding of the pathologies and more targeted therapies.

Authors should comment on the effect of these limitations on our understanding of cancer. These are limitations that certainly affect our understanding of the disease. Cancer is a complex, multifactorial process with molecular alterations and dysfunctions in multiple signaling and metabolic pathways. The lack of a comprehensive and dynamic view of protein interactions, post-translational modifications and metabolic contexts makes it difficult to identify the key mechanisms involved in cancer development. This lack of detailed affects cancer cell heterogeneity, variations in cellular responses, and difficulties in targeting specific proteins or metabolic pathways understanding.

The considerations raised about the complexity of protein interactions, post-translational modifications and metabolic contexts are relevant and could certainly affect the understanding and treatment of triple negative breast cancer and regulating immunity, biomarkers and potential immunotherapeutic.

We know triple negative breast cancer for its heterogeneity and lack of specific targets such as estrogen, progesterone and HER2 hormone receptors. This heterogeneity complicates therapy and can be influenced by multiple factors, including those I have mentioned, such as post-translational modifications of proteins and metabolic context. Detailed understanding of the metabolic pathways involved in this type of cancer could prove crucial for developing targeted therapies.

Regarding the regulation of immunity and immunotherapy, protein interactions and post-translational modifications can influence the body's immune response to cancer. Understanding the specific proteoforms involved in immunoregulation could be essential for developing effective immunotherapy strategies. As well as those involved in the control of immunity can act as targets for therapy and as biomarkers to monitor response to treatment.

An accurate understanding of the molecular mechanisms is essential to identify predictive biomarkers that allow predicting patient response to specific therapies. The spatio-temporal characteristics of proteins can influence these biomarkers and by complex interactions within the metabolic context.

In summary, taken together, these considerations are relevant and may profoundly influence the understanding and treatment of triple-negative breast cancer, as well as immunoregulation and immunotherapeutic potential. It is important to recognize that scientific research is constantly developing, and alternative approaches and technologies are emerging to address these challenges. Thus, while the limitations described may make understanding breast cancer complex, there is growing awareness of the challenges and efforts to overcome them through scientific innovation.

However, it is important to underline that cancer research is making significant progress thanks to the advancement of technologies and analytical approaches. Genomics, proteomics, metabolomics and other disciplines are contributing to a deeper understanding of the molecular alterations in cancers. Personalized and targeted therapies are emerging thanks to a better understanding of the specific molecular alterations that drive tumour growth. Systems biology and advanced technologies are offering a more dynamic and detailed view of protein interactions and metabolism.

The reader, beyond the mere description of metabolic facts, also wants to understand where these facts fit, with what limits and with what perspectives.

I ask the authors to add a paragraph that sets out and explains these considerations to the reader, obviously not as verbose as this referee did, but that makes simple sense.

Author Response

First of all, we would like to thank the reviewer for not only reviewing our manuscript, but also finding the review well-organized. 

Reviewer 2 Report

The manuscript by Das et al titled, “Extracellular vesicles in triple negative breast cancer: Immune regulation, biomarker, and immunotherapeutic potential”. Here are my suggestions as follows:

1.       Overall this is a very nicely written compact review. But the authors need to highlight the clinical, preclinical evidences to establish the translational significance of EV in a tabular manner.

2.       This manuscript completely missed the physiological significance of autophagic regulation concept of EVs and its better if a separate paragraph and a diagram is made for that.

3.       In Fig1, the lines appear to be vague. Its best if they are appropriately replaced with arrows as and when appropriate.

4.       In Fig2, the plant angle is not established in the figure. Legends need to be re-written.  Why the TNBC EVs represented in two different way?

5.       The authors should also dedicate a paragraph on the short comings of this field and highlight the challenges more vividly.

Author Response

(The authors gave the same response as above.)

Reviewer 3 Report

In the impressive review, the authors firstly give an overall introduction on EVs and their relationship with disease. Followingly, they provide a detailed pathophysiology and classification of TNBC and the in-depth understanding of the crucial roles of the EVs in the growth and metastatic dissemination of TNBC, its immune regulation, and their contribution as a predictive biomarker for TNBC. Finally, they focus on the recent key advances in immunotherapeutic strategies for better understanding the complex interplay between EVs and the immune system in TNBC and further developing EV-based targeted immunotherapies. Overall, this review is well organized on the summarization of EVs in immune regulation, biomarker, and immunotherapeutic potential for TNBC and should be published on Cancers except for some minor revisions.

In the section of EVs in TNBC immunotherapy, the authors mainly focus on the immune cell-derived EVs and self-loaded bioactive molecules as immunotherapeutics against TNBC. Other cells-derived EVs (especially TNBC tumor cell themselves) as the delivery platform or camouflage coating and loaded cargos such as immunomodulator and nanoparticles should also be fully summarized.

Author Response

First of all, we would like to thank the reviewer for providing critical comments for our manuscript.

Round 2

Reviewer 1 Report

I thank the authors for having clearly understood the spirit of my considerations and for having included them in the manuscript. Many authors do not even understand such considerations. Unfortunately, many do research with the logic of decades ago.

It is my opinion that the manuscript is now publishable

Author Response

We are so grateful for your time to review our manuscript and for your appreciation.

Best wishes

Reviewer 2 Report

Fig1A, pathway #6 that involves caspase according to the authors is not explained in the manuscript.

Rest of the clarifications are fine. Just that the autophagy part is written superficially without critical depth.

Author Response

Thank you very much for your suggestions. We have revised our manuscript with latest information about autophagy (Page3, Line no. 97-123) and also cite relevant papers.

Once again thank you for your efforts to improve our manuscript.

Round 3

Reviewer 2 Report

All comments have been clarified